# VENIS: Vision-centric Enhancement via Noise-Injection and Self-distillation for Multimodal Instruction Tuning

## Abstract

Multimodal Large Language Models (MLLMs) have shown great potential and broad prospects, but their instruction tuning faces critical challenges: the models pay insufficient attention to visual information and tend to prioritize learning textual content. This vision-deficient tendency directly leads to inadequate performance enhancement, weak ability to generalize across different scenarios, and frequent generation of hallucinatory content that deviates from visual facts. Existing solutions like expanding datasets or scaling architectures incur high costs with diminishing returns. This work introduces VENIS (**V**ision-centric **E**nhancement via **N**oise-**I**njection and **S**elf-distillation), a lightweight framework combining Noise-Injection and Self-distillation. It weakens textual priors by injecting random noise into instruction-response embeddings, forcing the model to ground its answers in visual information. Self-distillation then strengthens visual understanding while recovering textual knowledge. Experiments on LLaVA v1.5-7b and InternVL3-8B demonstrate consistent improvements across benchmarks. For LLaVA v1.5-7b, improvements include MMBench (+2.3%), MMVP (+7.4%), MMMU (+1.7%), and OCRBench (+1.6%). For InternVL3-8B, gains cover MMBench (+1.0%), MMMU (+3.1%), OCRBench (+4.6%), and HallusionBench (+1.7%). VENIS requires no additional data, annotations, or model modifications, offering an efficient reference for advancing multimodal instruction tuning.

## 1 Introduction

In recent years, Multimodal Large Language ModelsZhu et al. (2025); Bai et al. (2025) (MLLMs) have drawn increasing attention from both academia and industry, bridging the gap between visual perception and linguistic understanding. These models, which process both visual inputs and textual instructions to generate contextually coherent responses, have demonstrated remarkable capabilities across diverse tasks from image captioning and visual question answering (VQA) to complex multimodal reasoning. As the demand for AI systems that can seamlessly interpret and interact with the multimodal world grows, instruction tuning Liu et al. (2023) has become a critical step in refining MLLMs, enabling them to align their outputs with human intent and task-specific requirements. However, despite significant advancements, current instruction tuning paradigms are hindered by a fundamental flaw: MLLMs exhibit insufficient attention to visual information and a strong inclination toward learning textual content. This vision-centric deficiency manifests in limited performance gains on fine-grained visual tasks Chen et al. (2024); Tong et al., poor generalization across diverse scenarios Liu et al. (2024b), and persistent hallucinations Yifan Li & Wen (2023); Guan et al. (2024) where models generate responses that are semantically inconsistent with the visual content, exactly the core issues that our proposed Vision-centric Enhancement framework aims to address.

Existing approaches to address these issues often rely on resource-intensive strategies. For instance, enlarging datasets or refining annotations incurs substantial costs in data collection and curation, while replacing vision encoders Tong et al. (2024); Shi et al. (2025); Li et al. (2024) or scaling model architectures Bai et al. (2025); Zhu et al. (2025) leads to exponential increases in computational demands. Moreover, these methods frequently suffer from diminishing returns: beyond a certain point, adding more data or parameters yields marginal improvements, failing to fundamentally resolve the core issue of balancing reliance on visual content versus textual priors. This

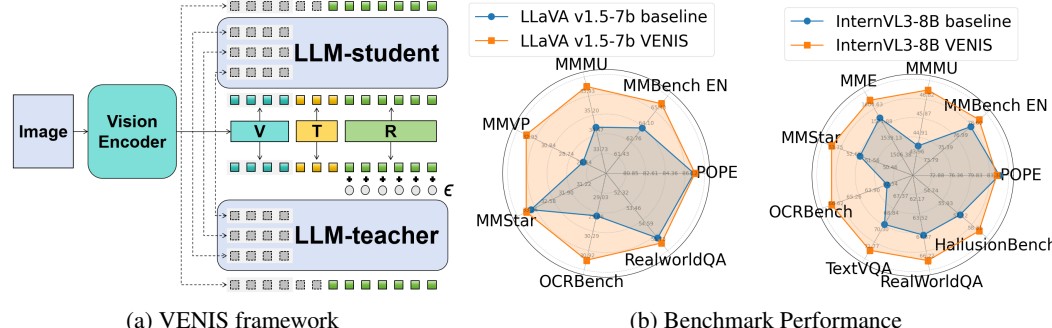

(a) VENIS framework          (b) Benchmark Performance

Figure 1: (a) VENIS framework, which includes a parallel student model processing original textual embeddings and a teacher model processing noisy textual embeddings, with teacher-student alignment of visual perception and understanding via Self-Distillation. (b) Radar chart showing the performance of LLaVA v1.5 Liu et al. (2024a) and InternVL3 Zhu et al. (2025) with/without VENIS, highlighting across-the-board gains.

over-reliance on textual patterns such as memorizing instruction-response pairs often causes models to "hallucinate" answers that align with linguistic expectations but ignore critical visual details, undermining their utility in real-world applications where grounding in visual reality is paramount.

To tackle these challenges without incurring prohibitive costs or modifying model architectures, we propose VENIS (**V**ision-centric **E**nhancement via **N**oise-**I**njection and **S**elf-distillation), a lightweight yet powerful framework for multimodal instruction tuning. VENIS operates on a simple yet effective insight: by deliberately weakening textual priors during training, the model is forced to prioritize visual information, while a subsequent self-distillation step ensures that critical linguistic knowledge is preserved. This dual-mechanism design addresses the trade-off between visual perception and textual proficiency, enhancing both without requiring additional data, annotations, or architectural changes.

The key contributions of this work are as follows:

- We propose a novel framework that integrates Noise-Injection and Self-Distillation. Noise-Injection disrupts textual embeddings to reduce reliance on linguistic patterns, while Self-Distillation aligns layer-wise visual features across teacher-student models, preserving visual understanding and recovering textual knowledge.

- In contrast to methods requiring massive data or enlarged architectures, VENIS eliminates the need for additional data, annotations, or model modifications. It imposes no inference costs and demands no adaptation efforts. The computational overhead is minimal: the Noise-Injection stage entails zero training costs, and the Self-Distillation stage adds only one extra LLM forward propagation during training. This modest investment is offset by substantial performance gains.

- Extensive experiments on two representative MLLMs, LLaVA v1.5-7b Liu et al. (2024a) and InternVL3-8B Zhu et al. (2025), demonstrate that VENIS consistently improves performance across diverse benchmarks, validating its generalizability to different model architectures. VENIS boosts key capabilities: LLaVA v1.5-7b gains on MMVP (+7.4%), MMBench (+2.3%), MMMU (+1.7%), and OCRBench (+1.6%); InternVL3-8B improves in MMBench (+1.0%), MMMU (+3.1%), OCRBench (+4.6%), and HallusionBench (+1.7%). These results collectively demonstrate VENIS's effectiveness, from strengthening visual perception to mitigating hallucinations.

## 2 RELATED WORK

In recent years, the field of multimodal large language models (MLLMs) has witnessed remarkable progress, with researchers striving to bridge the gap between visual perception and linguistic understanding.

Despite the progress made in MLLMs, instruction tuning, a crucial step in enhancing model performance, remains constrained by a core limitation: the models' inadequate focus on visual information and their pronounced tendency to prioritize textual learning. This imbalance manifests in lackluster performance improvements on visually grounded tasks, subpar generalization across diverse multimodal scenarios, and persistent hallucinations that diverge from visual realities. To enhance the visual capabilities of models, existing solutions fall into two main categories: data-centric and model-centric methods, each carrying distinct limitations.

## 2.1 Data-centric Methods: Increasing Data Quantity and Quality

High-quality visual instruction datasets are essential for achieving optimal performance in MLLMs. However, data construction is fraught with difficulties. Existing open-source instruction datasets Liu et al. (2024a); Dai et al. (2023); Zhu et al. (2024) often lack sufficient quantity, diversity, and creativity. Manual annotation is time-consuming and costly, while automated annotation methods cannot guarantee data quality and may increase the risk of hallucinations. Issues such as inaccurate, misaligned, or corrupted samples in pre-training data, and the potential for LLMs (e.g., GPT-4 OpenAI (2023)) to generate hallucinations when creating instruction tuning data, limit the effectiveness of data-centric approaches.

## 2.2 Model-centric Methods: Replacing Visual Encoders

The most commonly used visual encoders in MLLMs are Vision Transformers Dosovitskiy et al. (2021) trained with CLIP-style contrastive learning, with CLIP Radford et al. (2021) ViT-L being a popular architecture. However, these encoders have limitations. To address these, researchers have proposed various solutions, including replacing visual encoders with more diverse ones Li et al. (2024); Zhu et al. (2025); Bai et al. (2025), such as DINOv2 Oquab et al. (2023), SigLIP Zhai et al. (2023), and SigLIP2 Tschannen et al. (2025); using multiple visual encoders to leverage their respective advantages Tong et al. (2024); Shi et al. (2025); and aggregating features from different layers of visual encoders Cocchi et al. (2025) to enrich the visual input for LLMs. Nevertheless, replacing visual encoders introduces challenges including model compatibility issues, increased training costs, and higher technical complexity.

In summary, while existing methods for improving instruction tuning in MLLMs have made some progress, they are still plagued by limitations. Our proposed method, VENIS, aims to address these challenges by introducing a novel framework that does not rely on additional data, annotation, or major architectural changes, thereby providing a more efficient and effective solution for multimodal instruction tuning.

# 3 Methodology

This section elaborates on the proposed VENIS approach, aiming to provide a detailed and reproducible description to replicate the experiments. The overall architecture of VENIS is designed to enhance multimodal instruction tuning without modifying the vision encoder or model structure, and without the need for additional data or annotations.

## 3.1 Problem Formulation

Multimodal instruction tuning aims to enable the model to effectively understand and respond to multimodal instructions that include both visual and textual information. Specifically, the response generated by the model, denoted as $R$, is a text sequence where each element $R_t$ (representing the $t$-th token in the response) is predicted sequentially.

Formally, given a multimodal input consisting of a visual input $V$ and a textual instruction $T$, the model is tasked with generating a text sequence $R = [R_1, R_2, \ldots, R_N]$ (where $N$ is the length of the response sequence) such that each $R_t$ is consistent with the visual content in $V$, the semantic requirements of $T$, and the previously generated tokens $[R_1, R_2, \ldots, R_{t-1}]$.

Let $f_\theta(V, T, R_{1:t-1})$ represent the multimodal model with parameters $\theta$, which maps the input visual $V$, instruction $T$, and the preceding tokens $R_{1:t-1}$ to the probability distribution of the $t$-th token $R_t$.

The goal of instruction tuning is to optimize the parameters $\theta$ such that the generated sequence $R$ is as close as possible to the ground-truth sequence $R^* = [R_1^*, R_2^*, \ldots, R_N^*]$ in terms of semantic and contextual relevance. This is typically measured by maximizing the log-likelihood of the ground-truth sequence:

$$\max_\theta \sum_{t=1}^{N} \log P_\theta(R_t^* | V, T, R_{1:t-1}^*) \tag{1}$$

In the context of VENIS, the problem is further refined. We need to ensure that during the tuning process, when predicting each $R_t$, the model relies more on visual information $V$ while maintaining the necessary textual knowledge from $T$ and the coherence with the preceding tokens $R_{1:t-1}$. This requires addressing the issue where the model may over-rely on textual priors from $R_{1:t-1}$, leading to suboptimal utilization of visual content in $V$ and potential hallucinations in the generated $R_t$.

## 3.2 OVERALL ARCHITECTURE

The overall architecture of VENIS is designed to enhance the model's focus on visual information without modifying the model structure or adding additional data. It consists of two synergistic stages: Noise-Injection and Self-Distillation, as shown in Figure 1 (a). In the first stage, Noise-Injection, random noise is introduced into the embedding layer of the instruction-response text. This weakens the model's dependence on textual priors, forcing it to pay more attention to visual signals. In the second stage, Self-Distillation, the model aligns the visual feature outputs by each layer of the LLM during the process of understanding visual information, so as to recover the textual knowledge that may have been lost due to Noise-Injection and consolidate the visual understanding ability acquired in the first stage. The two stages work together to achieve the goal of enhancing visual-centric processing in multimodal instruction tuning, ensuring that the model can effectively utilize visual information while maintaining good performance in textual understanding and generation.

## 3.3 NOISE-INJECTION

The Noise-Injection step is designed to weaken the textual priors of the model, forcing the "multi-modal neurons" Goh et al. (2021) in the LLM to prioritize the use of visual signals.

### 3.3.1 MECHANISM

As pointed out in the related research, there are neurons in the LLM that can naturally handle multimodal semantics Schwettmann et al. (2023) and do not rely on the Projector to "translate" visual features into text-like embeddings Verma et al. (2024). When predicting token $R_t$, random noise added to textual embeddings of prior tokens $R_{1:t-1}$ prevents the LLM from relying solely on memorized textual patterns and forcing it to focus more on the visual pathway. Since the Projector does not encode high-level semantics, it is the layer-by-layer refine process of the LLM that can truly "understand" images Shukor & Cord (2024). Noise-Injection provides a stronger visual anchor for the layer-by-layer refine process of the LLM, allowing multimodal neurons to extract visual information during the layer-by-layer refinement.

### 3.3.2 IMPLEMENTATION DETAILS

For the textual embedding of the instruction-response pairs, we inject random Gaussian noise. Let $\text{emb}(T_R)$ be the original embedding of the instruction-response text, the noisy embedding $\text{emb}_{\text{noise}}(T_R)$ is calculated as:

$$\text{emb}_{\text{noise}}(T_R) = \text{emb}(T_R) + \epsilon \tag{2}$$

where $\epsilon \sim \mathcal{N}(0, \sigma^2)$ is the random Gaussian noise with mean 0 and variance $\sigma^2$. The value of $\sigma$ is a hyperparameter that can be adjusted based on specific tasks and datasets. Through experiments, we found that a $\sigma$ in the range of 0.01-0.001 generally yields good results.

## 3.4 SELF-DISTILLATION

The Self-Distillation step aims to maintain the focus on visual information while restoring the necessary textual knowledge, with special attention to aligning the LLM's layer-by-layer refine process of visual information understanding, using only visual features.

### 3.4.1 MECHANISM

Training with only noisy text can result in the loss of language priors and reduced generalization. By aligning the visual feature outputs of each layer through cosine similarity loss, the student model accurately replicates the teacher model's layer-by-layer refinement process of visual information. This ensures that the student model focuses exclusively on the visual pathway during distillation, avoiding interference from textual features. During the refine process, the student model not only consolidates the abstract semantics driven by vision but also supplements the language common sense masked by noise through the natural alignment ability of LLM. Thus, it achieves enhanced visual perception without language degradation, and does so without requiring new data, and ensures that the refinement of visual information in each layer of the LLM is effectively inherited and utilized.

### 3.4.2 IMPLEMENTATION DETAILS

We adopt single-stage, parallel teacher-student framework. During training, a "teacher" copy of the model receives noisy textual embeddings (as produced by the Noise-Injection step), while a "student" copy simultaneously receives the original, clean textual embeddings. The loss function consists of three components: cross-entropy loss for the teacher with respect to the ground-truth labels, cross-entropy loss for the student with respect to the ground-truth labels and visual feature distillation loss across layers. Firstly, there is the cross-entropy loss with respect to the ground-truth labels, $L_{\text{CE,student}}(\text{logits}_{\text{student}}, R^*)$ and $L_{\text{CE,teacher}}(\text{logits}_{\text{teacher}}, R^*)$, which ensures the basic generation quality of the response sequence. More importantly, to align the LLM's layer-by-layer refine process of visual information understanding, we perform feature distillation exclusively on the visual input $V$'s feature outputs of each layer of the LLM (excluding textual instruction $T$ and response $R$ features). Let $F_{V,\text{teacher}}^l$ be the visual feature output of the $l$-th layer of the teacher model, and $F_{V,\text{student}}^l$ be the visual feature output of the $l$-th layer of the student model. The loss for the $l$-th layer is calculated using cosine similarity:

$$L_{\text{cos}}^l = 1 - \cos(F_{V,\text{student}}^l, F_{V,\text{teacher}}^l) \tag{3}$$

where $\cos(\cdot, \cdot)$ denotes the cosine similarity function. The total visual feature distillation loss across all layers is the average of the losses from each layer:

$$L_{\text{feat}} = \frac{1}{L} \sum_{l=1}^{L} L_{\text{cos}}^l \tag{4}$$

where $L$ is the total number of layers in the LLM.

The total loss function for the student stage is:

$$L_{\text{total}} = L_{\text{CE,student}} + \alpha(L_{\text{CE,teacher}}) + L_{\text{feat}} \tag{5}$$

Here, $\alpha$ is a hyperparameter that balances the two cross entropy loss terms. Through experiments, we found that setting $\alpha$ in the range of 0.5 to 1 often leads to good performance, but it can be adjusted according to specific scenarios. This loss function ensures that the student model learns from the ground truth while strictly aligning with the teacher model in terms of the layer-by-layer processing of visual information, thereby better inheriting the teacher model's ability to understand visual content without being distracted by textual features.

### 3.5 COST ANALYSIS

VENIS avoids additional data collection or annotation, saving significant resources. The first stage (Noise-Injection) incurs no extra computational cost in training or inference compared to standard training.

For the second stage (Self-Distillation), a parallel teacher-student framework is used, with no additional time cost beyond combined training. However, it requires two LLM forward propagations to obtain layer-wise visual features for alignment, making total computational cost roughly twice that of a single training process. Notably, Self-Distillation adds minor training costs due to memory needs for feature alignment but no inference costs. Overall, VENIS maintains the original model's inference efficiency, with no extra structure or operations during inference.

## 4 EXPERIMENTS

In this section, we conduct extensive experiments to evaluate the effectiveness of VENIS on multimodal instruction tuning. The experiments are designed to address four core questions: its performance on multimodal tasks, cross-architecture consistency, component contributions, and impact on visual perception/language proficiency.

### 4.1 EXPERIMENTAL SETUP

#### 4.1.1 TRAINING DATASET

We use the llava-v1.5-mix665k Liu et al. (2024a) dataset (665k samples from COCO, GQA, etc.), chosen for its diverse multimodal tasks (captioning, VQA, reasoning) and strict quality control, facilitating fair comparison.

#### 4.1.2 PRE-TRAINED MODELS

We select two representative multimodal large language models as the base models for our experiments:

- **LLaVA v1.5** Liu et al. (2024a) A well-known open-source multimodal model that connects a vision encoder (CLIP ViT-L/14) with a language model (Vicuna-7b) via a projector. It has been pre-trained and fine-tuned on various multimodal datasets, providing a solid foundation for instruction tuning.
- **InternVL3-8B** Zhu et al. (2025) A state-of-the-art multimodal model with a powerful vision encoder and a language model, designed to handle diverse multimodal tasks. Its pre-trained weights InternVL3-8B-Pretrained are trained on large-scale image-text data, enabling it to capture rich visual and textual representations.

#### 4.1.3 EVALUATION BENCHMARKS

We use a diverse set of multimodal benchmarks, categorized into five types, to comprehensively evaluate the model's performance:

- **Comprehensive Multimodal** MMBench (fine-grained assessment) Liu et al. (2024b), MMVP (visual patterns) Tong et al., MMStar (visual dependency) Chen et al. (2024).
- **Multimodal Reasoning and Mathematics** MMMU (college-level tasks across 6 disciplines) Yue et al. (2024).
- **OCR, Chart, and Document Understanding** OCRBench (text recognition) Liu et al. (2024c), TextVQA (image text reasoning) Singh et al. (2019).
- **Real-World Comprehension** RealWorldQA (spatial/common sense understanding) xAI (2024).
- **Hallucination Evaluation** HallusionBench Guan et al. (2024), POPE (object hallucination) Yifan Li & Wen (2023).

#### 4.1.4 BASELINES

To ensure a rigorous and fair evaluation of VENIS, we define clear baselines to compare against.

- **LLaVA v1.5 baseline** We directly use the open-source LLaVA v1.5-7b weights as the baseline. These weights are the result of the original LLaVA v1.5 instruction tuning process, trained on the llava-v1.5-mix665k dataset using the official method.
- **InternVL3 baseline** This baseline is obtained by performing instruction tuning on the InternVL3-8B-Pretrained weights using the llava-v1.5-mix665k dataset, following the original instruction tuning method of InternVL3.

We compare the following variants to verify the effectiveness of VENIS:

|  | MMBench EN | MMMU | MMVP | MMStar | OCRBench | RealworldQA | POPE |
|---|---|---|---|---|---|---|---|
| LLaVA v1.5-7b | 63.8 | 34.6 | 27.3 | 33.4 | 29.7 | 55.9 | 86.9 |
| +VENIS | **66.1** | **36.3** | **34.7** | **33.6** | **31.3** | **56.3** | **87.0** |

Table 1: Performance comparison between the LLaVA v1.5-7b baseline and the model integrated with VENIS in 7 benchmark tests, demonstrating the improvement effect of VENIS on the core capabilities of this model

|  | MMBench EN | MMMU | MME | MMVP | MMStar | OCRBench | TextVQA | RealWorldQA | HallusionBench | POPE |
|---|---|---|---|---|---|---|---|---|---|---|
| InternVL3-8B | 78.4 | 44.2 | 1583 | **72.3** | 52.4 | 62.7 | 70 | 64.9 | 57.2 | **85.9** |
| +VENIS | **79.4** | **47.3** | **1621** | 70.3 | **54.3** | **67.3** | **72.5** | **66.9** | **58.9** | 85.4 |

Table 2: Performance of the InternVL3-8B baseline, which is derived by leveraging the InternVL3-8B-Pretrained pre-trained weights and applying the original InternVL3-8B instruction tuning method on the llava-v1.5-mix665k dataset, and the model integrated with VENIS in 10 benchmark tests, verifying the generalization of VENIS across different model architectures

- **Baseline** The aforementioned LLaVA v1.5-7b and InternVL3-8B baseline.
- **+VENIS** Adds Noise-Injection and Self-Distillation on top of the original instruction tuning pipeline.

### 4.1.5 IMPLEMENTATION DETAILS

Training uses batch size 128, learning rate 2e-5, 1 epoch (consistent with baselines). Noise-Injection adds Gaussian noise ($\mu = 0$, $\sigma = 0.01$) for the textual embedding of the instruction-response pairs. Self-Distillation employs a parallel teacher-student framework with loss components: student/teacher cross-entropy loss with a weighting coefficient ($\alpha = 0.5$), and layer-wise visual feature cosine loss. Experiments run on 8 NVIDIA H20 GPUs under identical environments.

### 4.2 MAIN RESULTS

We present the experimental results of VENIS on two representative MLLMs (LLaVA v1.5-7b and InternVL3-8B) across diverse benchmarks.

### 4.2.1 RESULTS ON LLAVA V1.5-7B

Table 1 summarizes the performance of LLaVA v1.5-7b and its variants on seven benchmarks, from which we can draw several key observations: +2.3% on MMBench EN (general multimodal capability), +1.7% on MMMU (reasoning), +7.4% on MMVP (visual detail perception), +0.4% on RealWorldQA (practical application), and +1.6% on OCRBench (OCR). These confirm enhanced visual perception and reasoning.

### 4.2.2 RESULTS ON INTERNVL3-8B

Table 2 presents the performance of InternVL3-8B and its variants across 10 benchmarks, from which we can draw the key observations: VENIS improves performance on 8/10 benchmarks, including +1.0% on MMBench EN, +3.1% on MMMU, +4.6% on OCRBench, and +1.7% on HallusionBench. This validates its cross-architecture generality.

### 4.2.3 SUMMARY OF MAIN RESULTS

In summary, the main results clearly demonstrate that VENIS consistently improves the performance of both LLaVA v1.5-7b and InternVL-8B across a wide range of benchmarks. The improvements are observed in various aspects, including general multimodal capabilities, OCR and text-related tasks, visual detail perception, multimodal reasoning, real-world application, and reduction of hallucinations. This validates that VENIS is an effective approach for enhancing the performance of MLLMs across different architectures.

| | MMBench EN | MMMU | MMVP | MMStar | OCRBench | RealworldQA | POPE |
|---|---|---|---|---|---|---|---|
| LLaVA v1.5-7b (Baseline) | 63.8 | 34.6 | 27.3 | 33.4 | 29.7 | 55.9 | 86.9 |
| +Noise-Injection | 65.1 | 35.4 | **35.3** | 33.3 | **31.4** | 56.2 | **87.3** |
| +Noise-Injection +Self-Distillation | **66.1** | **36.3** | 34.7 | **33.6** | 31.3 | **56.3** | 87.0 |

Table 3: Performance comparison among the LLaVA v1.5-7b baseline, the model with only Noise-Injection added, and the model with both Noise-Injection and Self-Distillation added in 7 benchmark tests, aiming to analyze the contribution of each component of VENIS

## 4.3 ABLATION STUDY

To dissect the contribution of each component in VENIS (i.e., Noise-Injection and Self-Distillation), we conduct an ablation study on LLaVA v1.5-7b. The performance of three model variants: baseline, baseline with only Noise-Injection, and baseline with both Noise-Injection and Self-Distillation compared across seven benchmarks, as shown in Table 3.

### 4.3.1 CONTRIBUTION OF NOISE-INJECTION

The introduction of Noise-Injection alone leads to notable improvements over the baseline across most benchmarks: +1.3% on MMBench EN, +0.8% on MMMU, +8.0% on MMVP, +1.7% on OCRBench, and +0.4% on POPE, highlighting its core role in enhancing general visual perception and understanding.

### 4.3.2 CONTRIBUTION OF SELF-DISTILLATION

When combining Noise-Injection with Self-Distillation, the model achieves further improvements or stabilizes performance compared to the Noise-Injection-only variant: +1.0% on MMBench EN, +0.9% on MMMU, and +0.1% on RealWorldQA, underscoring the role of Self-Distillation in recovering and refining knowledge.

### 4.3.3 SYNERGISTIC EFFECT

The ablation results confirm a synergistic relationship between the two components: Noise-Injection breaks over-reliance on textual priors to enforce visual perception and understanding, while Self-Distillation recovers critical textual knowledge and refines multimodal alignment. Together, they achieve the best performance across most benchmarks, with MMBench EN, MMMU, and RealworldQA showing cumulative gains that exceed the sum of individual component improvements. This synergy ensures that the model maintains strong visual focus while preserving language proficiency and domain adaptability, addressing the key challenge of balancing visual perception and knowledge retention in multimodal instruction tuning.

## 4.4 CASE STUDY

To clarify how VENIS enhances visual perception, we conducted layer-wise attention visualization on LLaVA v1.5-7b and its VENIS-enhanced variant, focusing on their attention distribution when processing a programming-related query from MMBench. We selected a sample from MMBench: an image showing three lines of results (True, False, False) with the query asking for the corresponding Python code (Options A-D). This task requires precise focus on visual details to match the correct code.

Figure 2 shows attention heatmaps for both the second layer and the final layer of the model. Attention visualization on a programming query from MMBench shows that, the baseline exhibits scattered attention, while related research has noted significant modality gaps in LLaVA's shallow layers Huang et al. (2024) (consistent with our observations here). In contrast, the VENIS models focus more on critical visual details (True/False results) in both shallow and final layers. This aligns with improved performance in visual-dependent tasks.

These results explain the MMBench performance gains for LLaVA v1.5. VENIS reduces reliance on distracting textual options, enabling accurate code-result matching by prioritizing key visual info.

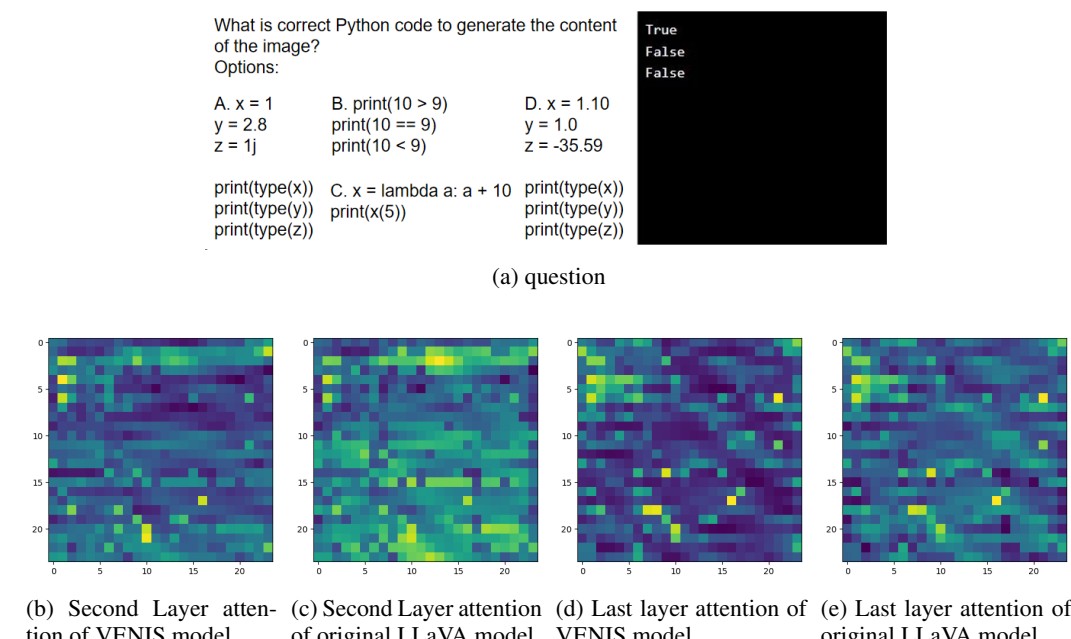

(a) question

(b) Second Layer attention of VENIS model.    (c) Second Layer attention of original LLaVA model.    (d) Last layer attention of VENIS model.    (e) Last layer attention of original LLaVA model.

Figure 2: Attention heatmaps of LLaVA v1.5 baseline vs. VENIS on an MMBench programming query. (a) Question; (b)(c) Second layer attention (VENIS/original); (d)(e) Last layer attention (VENIS/original). VENIS focuses more on critical details (True/False/False results) in all layers.

Its dual mechanisms (Noise-Injection and Self-Distillation) drive a shift to visual-centric processing, enhancing visual perception and multimodal understanding.

## 5 CONCLUSION

We propose VENIS, a lightweight approach enhancing multimodal instruction tuning via Noise-Injection and Self-Distillation. It weakens textual priors with Gaussian noise and aligns layer-wise visual features through self-distillation, forcing models to prioritize visual info while preserving language capabilities, without architectural changes or extra data. Experiments on LLaVA v1.5-7b and InternVL3-8B show consistent improvements in general multimodal capabilities, visual detail perception, OCR, reasoning, and hallucination reduction. The ablation study confirms the synergy of the two components. VENIS offers a practical reference for advancing multimodal instruction tuning, showing strategic visual-language alignment can drive MLLM progress.

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

# A APPENDIX

## A.1 STATEMENT ON LARGE-LANGUAGE-MODEL ASSISTANCE

In preparing this manuscript, the authors used a large-language model (LLM) solely for language polishing—i.e., grammar checks, word-choice refinement, and sentence-level stylistic adjustments. The LLM was not involved in study design, data collection or analysis, result interpretation, or the formulation of scientific arguments. All content remains the full responsibility of the authors.

