# OpenReview forum: "VENIS: Vision-centric Enhancement via Noise-Injection and Self-distillation for Multimodal Instruction Tuning"
_ICLR.cc/2026/Conference — ICLR 2026 Conference Withdrawn Submission_

### Official Review · Reviewer_aSkm · 2025-10-15

**Soundness:** 2
**Presentation:** 2
**Contribution:** 2
**Rating:** 2
**Confidence:** 3

**Summary:**

This paper introduces Vision-centric Enhancement via Noise-Injection and Self-distillation (VENIS), a lightweight framework to address the challenge of over-rely on textual information in MLLMS. The noise-injection are performed in the textual inputs, while the self-distillation is performed between the teacher with corrupted textual inputs and the student with clean textual inputs. Experiments on multiple benchmarks indicates the effectiveness of the proposed method.

**Strengths:**

1. The writting and the proposed method is easy to follow.
2. The proposed method can be easily integrated and its performance is good.

**Weaknesses:**

1. There lacks ablation study about the hyperparameters such as noise intensitiy.
2. From Table 3, seems the noise injection itself already brings observable performance gain. The authors should discuss this in more details.
3. Even though the proposed method seems to be effective, the authors do not explain the reasoning behind its effectiveness. Figure 2 does not provide much useful information for the interpretability.

**Questions:**

Please refer to the Weaknesses section.

---

### Official Review · Reviewer_R6xJ · 2025-10-27

**Soundness:** 1
**Presentation:** 2
**Contribution:** 1
**Rating:** 2
**Confidence:** 4

**Summary:**

The VENIS framework introduces a novel, cost-efficient tuning method that achieves consistent and significant performance gains across diverse benchmarks and model architectures.

**Strengths:**

The VENIS framework introduces a novel, cost-efficient tuning method that achieves consistent and significant performance gains across diverse benchmarks and model architectures (LLaVA v1.5-7b and InternVL3-8B), particularly in visual perception (MMVP: +7.4%) and hallucination mitigation

**Weaknesses:**

Fundamental Contradiction in Noise-Injection's Mechanism and Theoretical Justification : The central claim is that injecting noise into auto-regressive response embeddings ($R_{1:t-1}$) forces visual grounding. This is theoretically unsound. Disrupting the embeddings of preceding tokens in an LLM fundamentally destroys the necessary linguistic coherence and next-token prediction dependency, which is catastrophic for text generation. The observed gains are more likely a non-specific side effect of extreme regularization or data augmentation, rather than a reliable, theoretically-backed mechanism for "forced visual grounding," making the core narrative questionable.

Dilution of Language Knowledge and Inadequate Recovery via Visual-Exclusive Distillation : The framework admits that Noise-Injection leads to a "loss of language priors," but the proposed Self-Distillation only aligns the layer-wise visual feature outputs ($F_{V,student}^{l}$ and $F_{V,teacher}^{l}$) using a simple cosine similarity loss. This distillation target is strictly visual-exclusive. It provides zero explicit signal to recover the crucial high-level semantic, commonsense, or factual knowledge stripped by the text noise. Relying only on visual feature alignment is fundamentally inadequate to prevent language degradation, rendering the "preserving language capabilities" claim unsubstantiated.

Flawed Cost Analysis and Exaggerated Efficiency Claim : The paper claims "minimal" computational overhead and no additional training cost beyond the combined training time. However, the use of a parallel teacher-student setup mandates two full LLM forward propagations per training step to obtain the features for $L_{feat}$, effectively doubling the computational cost (FLOPs) and memory consumption for intermediate feature storage compared to the one-pass instruction tuning baseline. Misrepresenting a 100% training cost increase as "minimal" severely undermines the credibility of the efficiency claim.

Fragility and Instability of the Distillation Target from Noisy Input : The current self-distillation relies on the Teacher model (processing noisy text) to provide the target for the Student model (processing clean text). Why should the feature output of a model that has been intentionally handicapped by incoherent, noisy linguistic input serve as the ideal, enhanced visual alignment target? Features $F_{V,teacher}^{l}$ derived from corrupted context are inherently unstable and potentially suboptimal, indicating a significant flaw in the choice of distillation target.

Lack of Robustness: Primary Gain is Simple Regularization, Not Synergy : The ablation study shows that Noise-Injection alone provides the largest single-component gain on the core visual metric (MMVP jumps +8.0% to 35.3%) and that the addition of the complex Self-Distillation stage causes a drop in this key visual score (MMVP drops from 35.3% to 34.7%). This result directly contradicts the claimed "synergistic effect" and strongly suggests that the actual performance improvement is primarily driven by the non-specific regularization effect of text noise, rather than a successful theoretical mechanism for visual-language alignment.

Open Source Code Reproducibility:
The code for VENIS is currently available, but it does not provide explicit instructions or a tutorial on how to reproduce it. This lack of detail makes the experimental results hard to reproduce and verify independently.

**Questions:**

What is the theoretical and practical fallback solution for VENIS if a high-performing or fixed "Teacher" model is unavailable, or if the computational budget prohibits the use of the parallel teacher-student framework (i.e., running two full LLM forward passes) during training?

---

### Official Review · Reviewer_DzXo · 2025-10-30

**Soundness:** 2
**Presentation:** 3
**Contribution:** 2
**Rating:** 2
**Confidence:** 4

**Summary:**

The paper presents VENIS, a framework that boosts vision-centric reasoning in multimodal LLMs by (i) injecting Gaussian noise into the instruction/response embeddings to suppress spurious textual priors and (ii) performing layer-wise self-distillation on visual features to preserve language fluency. Experiments on LLaVA-v1.5-7B and InternVL3-8B with the 665 k-sample LLaVA-mix dataset report consistent gains on MMBench, MMVP, MMMU, OCRBench and HallusionBench without extra data, annotations or architectural changes.

**Strengths:**

The paper tries to address a impactful problem—vision under-reliance in multimodal instruction tuning—and proposes an refreshingly simple solution that requires no extra data, model surgery, or inference cost. The idea of corrupting textual embeddings to force visual grounding, then self-distilling only visual features, is interesting; the combination is well. Experiments cover two different architectures and a broad benchmark suite, accompanied by ablations.

**Weaknesses:**

**Scaling analysis is absent:** All tests are confined to 7–8 B parameters; without experiments on smaller (≤4 B) or larger (≥30 B) models the reader cannot judge whether the same noise/distillation recipe remains effective when the models to be trained are markedly weaker or stronger.

**Baseline comparison is thin:** The paper only contrasts VENIS with the authors’ own reproduction of vanilla instruction tuning. As a result, reported scores on MMMU, MMStar and OCRBench lag 15–25 points behind the official InternVL3 technical report. For example, the scores reported in this paper are 44.2 on MMMU, 52.4 on MMStar, and 62.7 on OCRBench, while the InternVL3 technical report reports 62.7, 68.2, and 88 on these benchmarks, respectively. I suggest that the authors provide the evaluation results of InternVL-8B-Pretrained. I suspect that the instruction tuning setting used in this paper does not bring performance gains to the pretrained model; therefore, the experimental improvements under this setting are insufficient to demonstrate the effectiveness of the proposed method.

**Minor LaTeX style issue:** Citations use \cite instead of \citep, producing “LLaVA v1.5 Liu et al. (2024a)” rather than “LLaVA v1.5 (Liu et al. 2024a)”. Please conform to standard ICLR format.

**Questions:**

Please see weaknesses.

---

### Official Review · Reviewer_DE3U · 2025-11-01

**Soundness:** 3
**Presentation:** 3
**Contribution:** 3
**Rating:** 4
**Confidence:** 5

**Summary:**

This submission introduces VENIS, a lightweight framework combining Noise Injection and Self-distillation. Noise Injection weakens textual priors by injecting random noise into instruction-response embeddings, forcing the model to ground its answers in visual information. Self-distillation then strengthens visual understanding while recovering textual knowledge. E

**Strengths:**

- The paper is well-motivated as focusing on the implicit influence caused by texture bias.

- It does not require new extra data and enlarge architecture, demonstrating the effectiveness of proposed approaches.

- It conducts experiments on two different backbones and brings little acceptable costs during training. The results show the consistent gains on various multimodal benchmarks while keeping the same training recipe with the baseline model.

**Weaknesses:**

- The main small weakness is the citation style used in this submission.

- The claim "random noise is introduced into the embedding layer of the instruction-response text. This weakens the model’s dependence on textual priors, forcing it to pay more attention to visual signals" seems no evidence (some experiments) to prove it. It is unknown what is behind it making it work. It is still unknown while simple Gaussian noise could influence the texture prior. How much the texture information is destroyed by noise should be studied.

- There is no evidence in submission to prevent the problem "self-distillation will bring the texture bias back." Although the teacher model in VENIS is trained with noisy textual embeddings to encourage stronger visual attention, this design may introduce a potential side effect. Since the teacher’s representations are partially degraded by noise, aligning the student’s features to those of the teacher through cosine similarity loss might lead the student to learn mean-level or smoothed representations rather than truly inheriting the visual enhancement. In other words, the distillation process could average out the noise effects instead of transferring the intended visual focus, potentially weakening the benefit of Noise-Injection.

- Table 1 and Table 2 should include the same test datasets.

- Table 3 shows that adding self-distillation brings little effects. It requires the in-depth analysis in Section 4.3 to explain while just noise-injection will bring the gains and destroy the text information in the early embedding layers. The author may require repeat experiments to remove the influence caused by performance fluctuation.

**Questions:**

See weakness.
I would like to raise my score if the main technique concerns in the weakness section are addressed.

---

### Note · Authors · 2025-11-12

I have read and agree with the venue's withdrawal policy on behalf of myself and my co-authors.